# Comparison of Satellite-Based Sea Surface Temperature to In Situ Observations Surrounding Coral Reefs in La Parguera, Puerto Rico

**Andrea M. Gomez** [1,2,3,*], **Kyle C. McDonald** [1,2,3,4], **Karsten Shein** [5], **Stephanie DeVries** [6], **Roy A. Armstrong** [7], **William J. Hernandez** [2] **and Milton Carlo** [7]

1 Ecosystem Science Lab, Department of Earth and Atmospheric Sciences, The City College of New York, 160 Convent Ave, New York, NY 10031, USA; kmcdonald2@ccny.cuny.edu
2 NOAA CESSRST, City College of New York, New York, NY 10031, USA; william.hernandez@upr.edu
3 Earth and Environmental Sciences Program, The Graduate Center, City University of New York, New York, NY 10016, USA
4 Carbon Cycle and Ecosystems Group, Jet Propulsion Laboratory, California Institute of Technology, 4800 Oak Grove Drive, Pasadena, CA 91001, USA
5 ExplorEIS, Ashville, NC 28801, USA; karsten@exploreis.com
6 Biology, Geology & Environmental Science, University of Tennessee Chattanooga, Chattanooga, TN 37403, USA; stephanieldevries@gmail.com
7 Department of Marine Sciences, University of Puerto Rico, Mayagüez, PR 006682, USA; roy.armstrong@upr.edu (R.A.A.); mcarlo@hotmail.com (M.C.)
* Correspondence: Agomez@gradcenter.cuny.edu

**Abstract:** Coral reefs are among the most biologically diverse ecosystems on Earth. In the last few decades, a combination of stressors has produced significant declines in reef expanse, with declining reef health attributed largely to thermal stresses. We investigated the correspondence between time-series satellite remote sensing-based sea surface temperature (SST) datasets and ocean temperature monitored in situ at depth in coral reefs near La Parguera, Puerto Rico. In situ temperature data were collected for Cayo Enrique and Cayo Mario, San Cristobal, and Margarita Reef. The three satellite-based SST datasets evaluated were NOAA's Coral Reef Watch (CoralTemp), the UK Meteorological Office's Operational SST and Sea Ice Analysis (OSTIA), and NASA's Jet Propulsion Laboratory (G1SST). All three satellite-based SST datasets assessed displayed a strong positive correlation (>0.91) with the in situ temperature measurements. However, all SST datasets underestimated the temperature, compared with the in situ measurements. A linear regression model using the SST datasets as the predictor for the in situ measurements produced an overall offset of ~1 °C for all three SST datasets. These results support the use of all three SST datasets, after offset correction, to represent the temperature regime at the depth of the corals in La Parguera, Puerto Rico.

**Keywords:** satellite SST; in situ; coral reefs

## 1. Introduction

In the last few decades, a combination of biotic and abiotic stressors has resulted in significant declines in reef expanse and diversity. Climate change and associated increasing ocean temperatures have resulted in heat stress being identified as one of the greatest threats to coral reef health [1–3]. Tropical shallow water reef building corals live near the upper limit of their thermal tolerance, and a temperature change of as little as 1–2 °C can be detrimental to their health [4]. Further, zooxanthellae experience photoinibition as a result of elevated temperature and light exposure, which damage their photosynthetic

systems [5]. Coral bleaching by heat stress involves the production of excess reactive toxic oxygen species, which contribute to oxidative damage and lead to metabolic dysfunction, and sometimes the expulsion of the symbiotic zooxanthellae (i.e., bleaching). Depending on the duration of the event, heat stress can ultimately cause coral death [6–8]. In October 2015, the US National Oceanic and Atmospheric Administration (NOAA) declared the third global coral bleaching event was underway. That global event ended in May 2017, but not before affecting more coral reefs worldwide than previously documented bleaching events, and causing record thermal stress in some areas that had never experienced mass bleaching [9]. Unusually warm sea surface temperatures in the Atlantic were also one of the driving factors for the active 2017 Atlantic hurricane season [10], with some of Puerto Rico's coral reefs being extensively physically damaged by the transit of multiple major hurricanes during the season [11].

The local monitoring of coral reefs by snorkeling or scuba diving provides important detailed information regarding reef health at local scales, but resource limitations restrict the coverage and repeatability of such monitoring to a small proportion of coral reefs globally. The ability to utilize remote sensing techniques to survey corals on broader geographic scales is therefore critical for assessing the effects of anthropogenic climate change in remote or inaccessible areas. Efforts to monitor coral reef environmental conditions in near-real-time on broader (e.g., regional or global) scales currently rely on satellites, because extensive in situ surveys can be cost and time-prohibitive [12]. However, in situ observations at the surface of the ocean, as well at the depth of the corals, are needed to evaluate and improve the accuracy of remote sensing datasets, especially in the shallow, near-shore reef zone, where adjacent land and highly variable benthic albedo can introduce bias in satellite-based measurements [13]. With current technology, the health of coral reef ecosystems cannot be directly observed by satellites in Earth orbit, however, satellite-derived sea surface temperature (SST) data can serve as a proxy for predicting where and when heat stress events can lead to coral bleaching [14].

NOAA's Coral Reef Watch (CRW) program has developed a suite of near real-time satellite SST-based products to monitor heat stress on coral reefs worldwide. CRW SST-based products (Versions 2.0 and 3.0) were used extensively to monitor and document the third global coral bleaching event [9]. CRW calculates the thermal stress for each reef location that can lead to coral bleaching, by comparing near real-time SST values with a long-term SST climatology. The SST climatology supporting CRW's current version 3.1 daily global 5 km product suite is derived from a combination of NOAA/National Environmental Satellite, Data, and Information Service (NESDIS) 2002–2012 reprocessed daily global 5 km Geo-Polar Blended Night-only SST Analysis, and the 1985–2002 daily global 5 km nighttime SST reanalysis, produced by the United Kingdom (UK) Metrological Office, on the Operational SST and Sea Ice Analysis (OSTIA) system. CRW's suite of 5 km products includes SST, SST Anomaly, Coral Bleaching HotSpot, Degree Heating Week (DHW), a 7-day maximum Bleaching Alert Area, and a 7-day SST trend [15]. CRW satellite SST datasets consist of night-only temperature measurements, because this helps to reduce the diurnal temperature fluctuation biases that would occur if both day and night measurements were used [16]. Previous research has also found that in the tropics, night-only temperature measurements agreed more favorably with in situ buoys at 1 m depths [17].

While gridded SST satellite products are usually adequate for monitoring offshore and large spatial areas, the same measurements may not be representative for coral reef ecosystems found in shallow coastal waters [18–20]. A number of previous studies comparing other remote sensing SST datasets and in situ temperature measurements at different geographical locations have found significant differences between these measurements [21–23]. Such discrepancies are caused by a combination of factors, including coarse satellite spatial and temporal resolutions, contamination of the satellite footprint by land areas, and the complexity of the environment in coastal zones (e.g., ocean mixing, increased turbidity, dissolved organic compounds from land [24,25]).

Since coral reef ecosystems are found along coastlines and many coral reef managers use CRW's SST-based products to monitor reefs, it is important to understand the accuracy of these CRW SST

products in specific locations, as well as to establish that the SST datasets are optimal [26,27]. Currently, there are no case studies examining the applicability of CRW's (Version 3.1) daily global 5 km satellite SST product of in situ temperature measurements at the depth of the corals in La Parguera, Puerto Rico. To help address this challenge and ensure that CRW's SST products are optimally suited for assessing the temperature at the depth of coral reefs, we deployed a network of in situ temperature sensors at varying depths across four coral reefs at La Parguera, Puerto Rico. Data collected from this network were used as a case study to assess correspondence between CRW's 5 km satellite-based SST product [15], the Jet Propulsion Laboratory's (JPL) 1 km SST dataset [28], and the UK Metrological Office's (OSTIA) 5 km SST product [29]. The CRW satellite dataset is daily, global, and nighttime-only at 5 km (0.05° latitude/longitude) resolution. The OSTIA SST and G1SST are both global daily datasets and support comparison with the CRW SST.

The goals of this case study were to:

1.　Assess temperature representation at different coral depths and locations in La Parguera, Puerto Rico, by the three satellite-based remote sensing SST datasets.
2.　Identify a statistical model that predicts coral depth temperature from the remote sensing SST datasets.

## 2. Materials and Methods

### 2.1. Study Area

La Parguera, located in southwestern Puerto Rico, was selected for the in situ temperature logger deployment (Figure 1), in part because these reefs are extensively environmentally and biologically monitored [30–32], and because NOAA CRW's satellite SST products lack validation datasets in this region. According to paleoclimate data obtained from a coral core, this area has experienced a 2 °C increase between 1751 and 2004 [33]. The local average SST is 27.9 °C, with an annual variability of 3.2 °C (derived from daily measurements between 1966 and 2002 [34]). SST near southwest Puerto Rico are influenced by continental freshwater runoff from the Orinoco and Amazon rivers [35]. This area was affected by the third global bleaching event and the active 2017 Atlantic hurricane season [36,37]. However, the damage from bleaching and hurricanes was non-uniform, and minimal in certain locations.

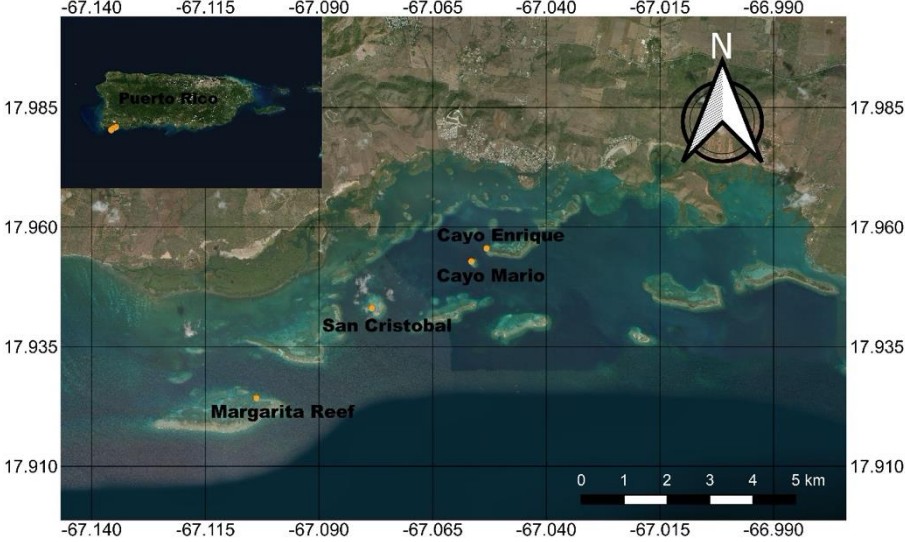

**Figure 1.** Map of study location, La Parguera, Puerto Rico. During June 2017, six temperature sensors were deployed at Cayo Enrique, and six temperature sensors at Cayo Mario (*n* = 12 total). The depths ranged from 1–13 m. In March 2019, eight temperature sensors were deployed at San Cristobal between depths of 5–6 m, and another eight were deployed at Margarita Reef at depths between 3–4 m (*n* = 16).

The reef system in La Parguera consists of both nearshore and offshore reefs, and is mostly composed of fringing, bank barrier reefs, and submerged patch reefs. The reefs here are experiencing stress from increasing ocean temperatures, rapid coastal development producing an influx of sediments and nutrients, and physical impacts from transitory tropical storms [38].

*2.2. In Situ Temperature Loggers*

During June 2017, we deployed a total of 12 temperature data loggers in situ within the coral reefs using SCUBA. Six data loggers were installed at varying depths (ranging from 1 m–13 m) at Cayo Enrique (17.95554 N, −67.05312 W), and six loggers at Cayo Mario (17.95283 N, −67.05648 W). The loggers used were HOBO Pendant 64 K Temperature and Light pendant loggers (Onset Computer Corporation, Massachusetts, U.S.A.; ±0.53 °C accuracy at 27.6 °C), and were chosen because HOBO loggers are widely used in coral studies [19,39–44]. These loggers were set to record water temperature at 15-min sampling intervals. Before deployment, the loggers were calibrated against a HOBO Water Temperature Pro v2 Data logger (±.21 °C accuracy at 27.6 °C) for 20 h, to establish relative baseline accuracy and ensure comparability.

The CRW satellite SST dataset was the primary focus of comparison, since it is used by coral mangers to monitor corals. Therefore, in March 2019, an additional 16 in situ temperature loggers were deployed at two new reef locations. This allowed comparison of three, CRW 5 km pixel cells, rather than one; Cayo Enrique and Cayo Mario are situated in one, 5 km pixel. Eight temperature loggers were deployed at San Cristobal (17.94302 N, −67.07834 W) at depths between 5–6 m, and eight temperature loggers at Margarita Reef (17.92422 N, −67.10377 W) at depths around 3–4 m. The temperature loggers used were the same ones deployed at Cayo Enrique and Cayo Mario, with the identical settings. They were calibrated in an ice bath before deployment, to ensure they were functioning within their manufactured accuracy (±0.53 °C accuracy). Table 1 lists information for the depth and location of each logger deployed at all the reef sites.

**Table 1.** Deployment depth of loggers and associated temperature offset from the calibration.

| Location | Logger # | Depth (m) | Offset (°C) |
|---|---|---|---|
| Cayo Enrique | 1 | 10 | −0.00759 |
| | 2 | 6 | 0.081311 |
| | 3 | 5 | 0.223544 |
| | 4 | 10 | 0.189653 |
| | 5 | 10 | 0.151872 |
| | 6 | 6 | 0.191875 |
| Cayo Mario | 7 | 13 | 0.195092 |
| | 8 | 11 | 0.160089 |
| | 9 | 3 | 0.173979 |
| | 10 | 11 | 0.137865 |
| | 11 | 3 | 0.020634 |
| | 12 | 1 | 0.083416 |
| San Cristobal | 1 | 5 | 0.343 |
| | 2 | 5 | 0.232 |
| | 3 | 5 | 0.343 |
| | 4 | 5 | 0.232 |
| | 5 | 5 | 0.232 |
| | 6 | 6 | 0.232 |
| | 7 | 6 | 0.232 |
| | 8 | 6 | 0.343 |
| Margarita Reef | 1 | 3 | 0.232 |
| | 2 | 3 | 0.232 |
| | 3 | 3 | 0.232 |
| | 4 | 4 | 0.232 |
| | 5 | 4 | 0.232 |
| | 6 | 4 | 0.343 |
| | 7 | 4 | 0.343 |
| | 8 | 4 | 0.232 |

The loggers were placed on or near the coral reef framework and secured with zip ties, to minimize movement in the water. Data retrieval was performed in situ, using an optical data shuttle to minimize the potential for observational interruptions, data loss, and instrument disturbances. Night-time observations were extracted for analysis from the time series logger data to compare to CRW's night-time only SST. We evaluated the in situ time series and determined that, owing to the short duration of the study and the use of night-time only observations, drift corrections were unnecessary.

## 2.3. Satellite Sea Surface Temperature Datasets

The three different gridded, global, daily remote sensing SST datasets listed in Table 2 were assessed in this study. The NOAA CRW program's Version 3.1 products are based on CRW's CoralTemp Version 1.0 SST dataset and are derived from NOAA/NESDIS' operational near-real-time daily global 5 km Geostationary-Polar-orbiting Blended Night-only SST Analysis from late 2016 onwards. OSTIA is also generated globally, and daily, at 5 km resolution, and has been found to be the most suitable SST dataset for aquaculture studies [23]. The SST measurements used in OSTIA are provided by the Group for High Resolution SST, which uses a combination of in situ and satellite data from both infrared and microwave radiometers to generate the SST [29]. The Group for High Resolution Sea Surface Temperature (GHRSST) daily, global 1 km SST (G1SST) dataset is produced by the JPL Regional Ocean Modeling System group. This product is created by using a multi-scale two-dimensional variational blending algorithm on a global 0.0009 degree grid, and is also a combination of data from multiple satellites and in situ data [28]. Remote sensing SST values nearest to the in situ sites were extracted from netCDF4 files by means of a spherical nearest neighbor approach, limited to open water vectors.

**Table 2.** Remote sensing datasets evaluated in this study.

| SST Datasets | Source | Resolution Spatial | Resolution Temporal | Day/Night | References |
|---|---|---|---|---|---|
| CoralTemp | NOAA/NESDIS CRW | 5 km | Daily | Night | [15] |
| OSTIA | UKMO | 5 km | Daily | Day + Night | [28] |
| G1SST | JPL ROMS | 1 km | Daily | Day + Night | [29] |

## 2.4. In Situ and Satellite SST Statistical Comparisons

To facilitate comparison with the three remote sensing SST datasets, in situ temperature observations between 19:00 and 06:00 (local time sunset/sunrise) were extracted and averaged to produce a daily nighttime-only in situ data series. A Spearman Correlation was performed on the in situ temperature data at each reef site to compare the data from the different logger sites and depths. No statistical difference was found among sites or depth ($p < 0.05$; $r > 0.83$ for all logger sites), indicating a strong correlation between loggers at each site. Therefore, for the statistical computations with the remote sensing SST datasets, the in situ loggers were averaged, to yield one daily, in situ measurement time series. Another reason we decided to average the in situ loggers to produce one daily in situ measurement for each site ($n = 3$), was because some temperature loggers were lost during the course of the study due to hurricanes, earthquakes, and the harsh ocean environment, and of the remaining data loggers, not all were able to capture data by 1 March 2020, when the study ended. On the conclusion of the study in early March 2020, only two loggers were recovered from Cayo Enrique, three from Cayo Mario, three from Margarita Reef, and five from San Cristobal.

For the duration of the study, the daily minimum, maximum, and mean temperatures were computed, along with their standard deviations for Cayo Enrique and Cayo Mario (30 June 2017–1 March 2020), San Cristobal (1 March 2019–1 March 2020), and Margarita Reef (1 March 2019–24 December 2019). The bias (satellite—in situ data) was also calculated for each satellite-based SST dataset, to relate the daily differences between the satellite and in situ temperature logger measurements. A mean bias value above zero corresponds to cooler SST data than the in situ data, and a bias below zero stands for warmer SST data [45]. From the biases, the mean, standard deviation, and root mean square error were

calculated. Next, a Spearman's correlation was performed to evaluate the strength of the relationship between the SST datasets and in situ temperature measurements. The correlation coefficient from the test varies between +1 and −1, with values closer to +1, indicating a stronger degree of association between the measurements [46].

Scatter plots of the in situ temperature measurements against the satellite SST were also produced to evaluate the relationship. Finally, a linear regression model was fit to adjust for differences between the remote sensing SST datasets and in situ measurements, with the satellite SST datasets as the predictor for the averaged in situ temperature recorded by the loggers. Given the close, linear agreement between the two variables as observed in the scatter plots, the linear regression model, while not accounting for serial correlation in the residuals, satisfies the goals of this work to fit a model that can effectively estimate reef-depth water temperature from SST estimated by satellite remote sensing.

## 3. Results

For Cayo Enrique and Cayo Mario, San Cristobal, and Margarita Reef, summaries of the minimum, maximum, mean, and standard deviation (SD) for the averaged in situ temperature and three remote sensing SST datasets for the duration of the study are evaluated, and summarized in Tables 3–5. The mean bias (°C), its SD, and the root mean square error (RMSE) for each remote sensing SST dataset, along with the Spearman correlation coefficient, are summarized for all locations in Tables 6–8 For JPL's Level 4, daily, G1SST dataset, after 9 December 2019 to the present, the SST dataset has been reported to produce poor quality SST data, and this poor data was removed from the analysis (https://podaac.jpl.nasa.gov/announcements/2020-01-29_G1SST_Data_Outage_Alert).

**Table 3.** Summary of the sea surface temperature (SST) data (°C) for Cayo Enrique and Cayo Mario from 30 June 2017–1 March 2020 (~two years and eight months; *n* = 975 days) for the averaged in situ loggers (data gap between 11 September 2019–1 October 2019), CoralTemp, and Operational SST and Sea Ice Analysis (OSTIA). The G1SST dataset is from 30 June 2017–8 December 2019 (*n* = 891 days).

| SST Source | Min | Max | Mean | SD |
|---|---|---|---|---|
| **In situ** | 26.23 | 30.47 | 28.41 | 1.13 |
| CoralTemp | 26.21 | 30.24 | 28.09 | 1.02 |
| OSTIA | 26.21 | 30.15 | 28.15 | 1.01 |
| G1SST | 25.44 | 30.31 | 28.21 | 1.08 |

**Table 4.** Summary of the SST data (°C) for San Cristobal from 1 March 2019–1 March 2020 (one year) for the averaged in situ loggers, CoralTemp, and OSTIA. The G1SST dataset is only from 1 March 2019–8 December 2019.

| SST Source | Min | Max | Mean | SD |
|---|---|---|---|---|
| In situ | 26.13 | 30.24 | 28.37 | 1.09 |
| CoralTemp | 26.28 | 30.20 | 28.22 | 1.03 |
| OSTIA | 26.25 | 30.17 | 28.25 | 1.03 |
| G1SST | 26.14 | 30.35 | 28.57 | 0.97 |

**Table 5.** Summary of the SST data (°C) for Margarita Reef from 1 March 2019–24 December 2019 (~10 months) for the averaged in situ loggers, CoralTemp, and OSTIA. The G1SST dataset is only from 1 March 2019–8 December 2019.

| SST Source | Min | Max | Mean | SD |
|---|---|---|---|---|
| In situ | 26.27 | 30.23 | 28.55 | 1.02 |
| CoralTemp | 26.27 | 30.19 | 28.43 | 1.02 |
| OSTIA | 26.27 | 30.14 | 28.48 | 1.02 |
| G1SST | 26.23 | 30.82 | 28.55 | 0.96 |

**Table 6.** Mean bias (satellite—in situ), standard deviation of the bias, root mean square error (RMSE) of the bias, and Spearman correlations between satellite-based SST datasets and averaged in situ temperature for Cayo Enrique and Cayo Mario from 30 June 2017–8 December 2019 (about two years and five months; *n* = 891 days).

| SST Dataset | Mean Bias (°C) | SD of the Bias | RMSE | Spearman |
| --- | --- | --- | --- | --- |
| CoralTemp | −0.35 | 0.38 | 0.52 | 0.94 |
| OSTIA | −0.29 | 0.37 | 0.47 | 0.93 |
| G1SST | −0.33 | 0.40 | 0.52 | 0.92 |

**Table 7.** Mean bias (satellite—in situ), standard deviation of the bias, root mean square error (RMSE) of the bias, and Spearman correlations between satellite-based SST datasets and averaged in situ temperature for San Cristobal from 1 March 2019–1 March 2020 (one year) for CoralTemp, and OSTIA datasets. The G1SST dataset is only from 1 March 2019–8 December 2019.

| SST Dataset | Mean Bias (°C) | SD of the Bias | RMSE | Spearman |
| --- | --- | --- | --- | --- |
| CoralTemp | −0.16 | 0.33 | 0.36 | 0.95 |
| OSTIA | −0.12 | 0.30 | 0.33 | 0.96 |
| G1SST | −0.11 | 0.31 | 0.33 | 0.91 |

**Table 8.** Mean bias (satellite—in situ), standard deviation of the bias, root mean square error (RMSE) of the bias, and Spearman correlations between satellite-based SST datasets and averaged in situ temperature for Margarita Reef from 1 March 2019–24 December 2019 (~10 months) for CoralTemp, and OSTIA datasets. The G1SST dataset is only from 1 March 2019–8 December 2019.

| SST Dataset | Mean Bias (°C) | SD of the Bias | RMSE | Spearman |
| --- | --- | --- | --- | --- |
| CoralTemp | −0.12 | 0.30 | 0.32 | 0.95 |
| OSTIA | −0.07 | 0.29 | 0.30 | 0.95 |
| G1SST | −0.05 | 0.35 | 0.36 | 0.91 |

All the remote sensing SST datasets evaluated yielded similar temperature patterns, and correspondingly high correlations with the in situ temperature measurements (correlation coefficients >0.91), and the seasonal trends can be observed in the time series (Figures 2–4). A strong seasonal trend is observed, with the three remote sensing datasets consistently underestimating the temperature at the depth of the corals during warmer months (June to September).

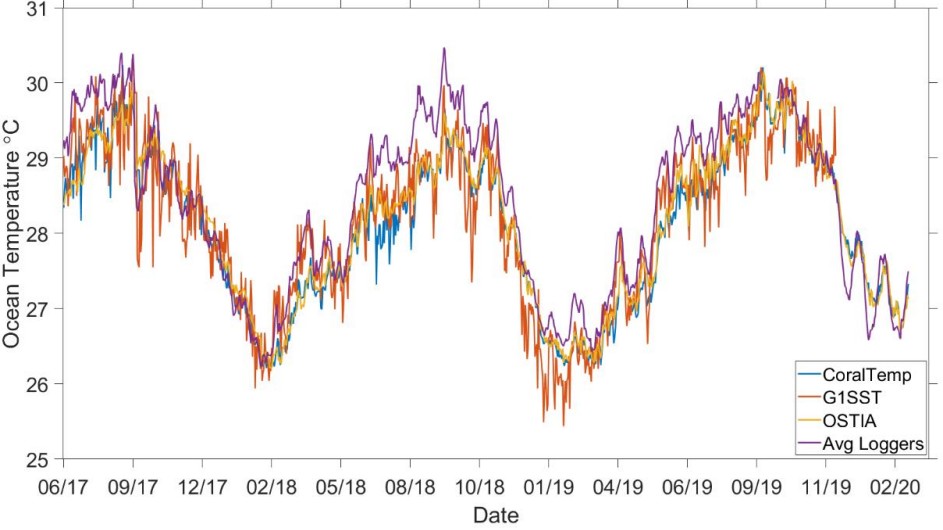

**Figure 2.** Time series of temperature data from 30 June 2017–1 March 2020 of the three remote sensing datasets (CoralTemp, G1SST, OSTIA) and the averaged in situ loggers for Cayo Enrique and Cayo Mario.

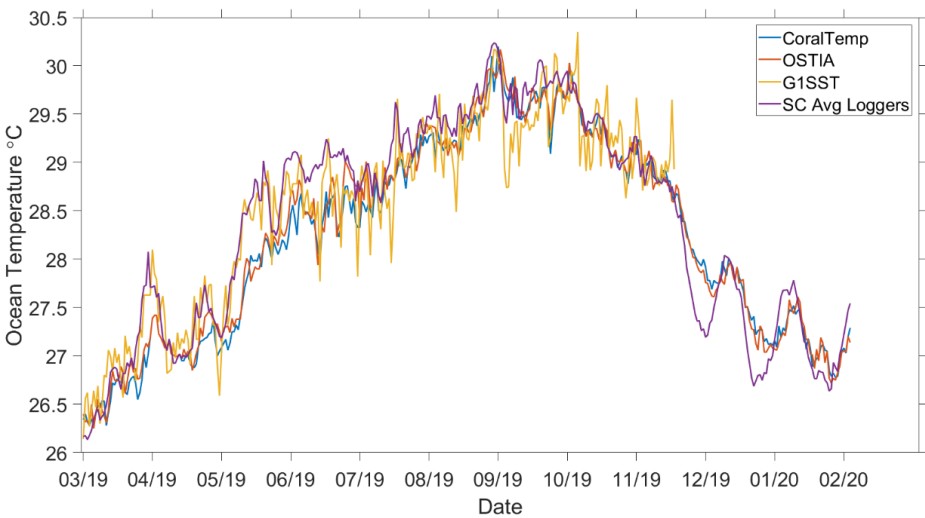

**Figure 3.** Time series of temperature data from 1 March 2019–1 March 2020 of the three remote sensing datasets (CoralTemp, G1SST, OSTIA), and the averaged in situ loggers for San Cristobal.

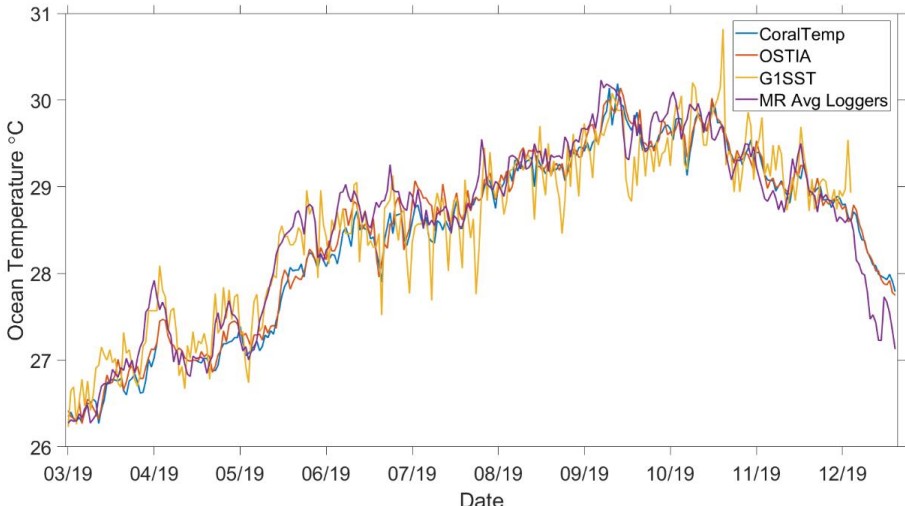

**Figure 4.** Time series of temperature data from 1 March 2019–24 December 2019 of the three remote sensing datasets (CoralTemp, G1SST, and OSTIA), and the averaged in situ loggers for Margarita Reef.

Ocean temperature seasonality was also explored for Cayo Enrique and Cayo Mario for only two years of the study (30 June 2017–30 June 2019), and it was found that correlation coefficients decreased marginally, but not significantly, when split into the dry and wet seasons. The first dry season (December 2017 to March 2018) had correlation coefficients of 0.90, 0.89, and 0.87 for CoralTemp, G1SST, and OSTIA, respectively. The wet season (April 2018 to November 2018) had slightly higher coefficients of 0.92, 0.89, and 0.95 for CoralTemp, G1SST, and OSTIA, respectively. Overall, all remote sensing SST datasets displayed negative (cool) biases for the majority of the study, and this can be observed in Figures 5–7.

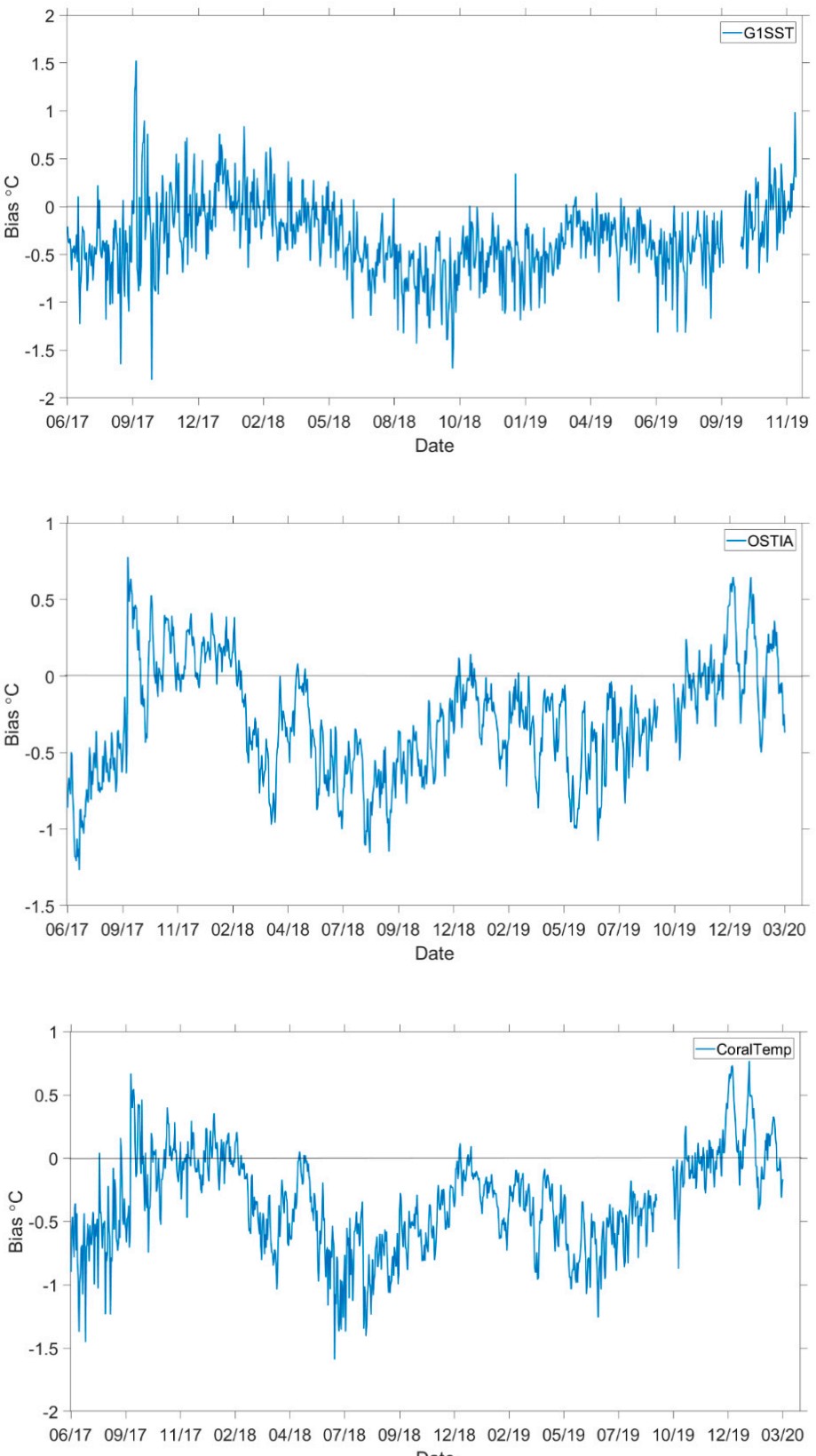

**Figure 5.** Time series of the bias (satellite—in situ) temperatures from 30 June 2017–1 March 2020 for Cayo Enrique and Cayo Mario.

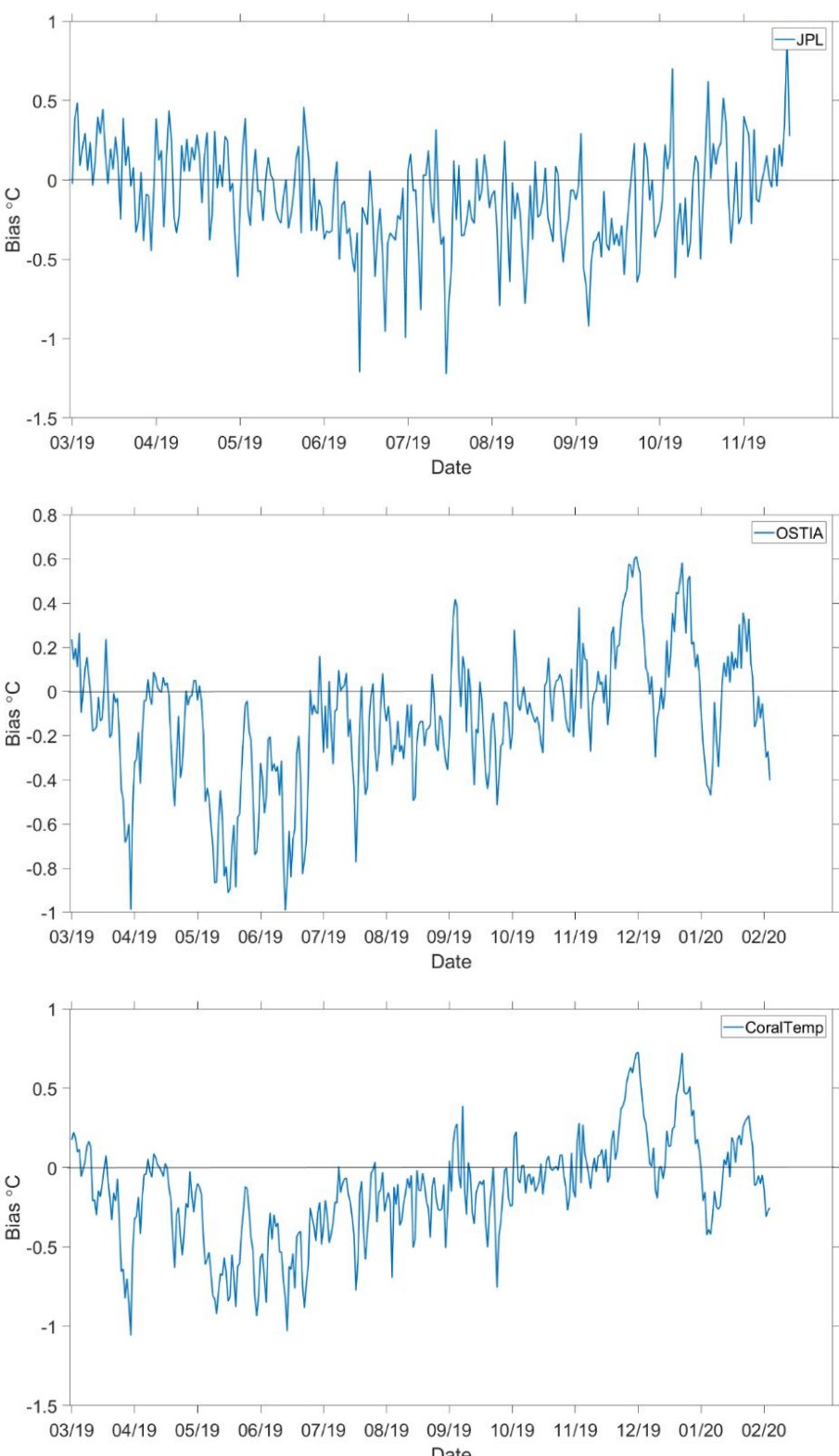

**Figure 6.** Time series of the bias (satellite—in situ) temperatures from 1 March 2019–1 March 2020 for San Cristobal.

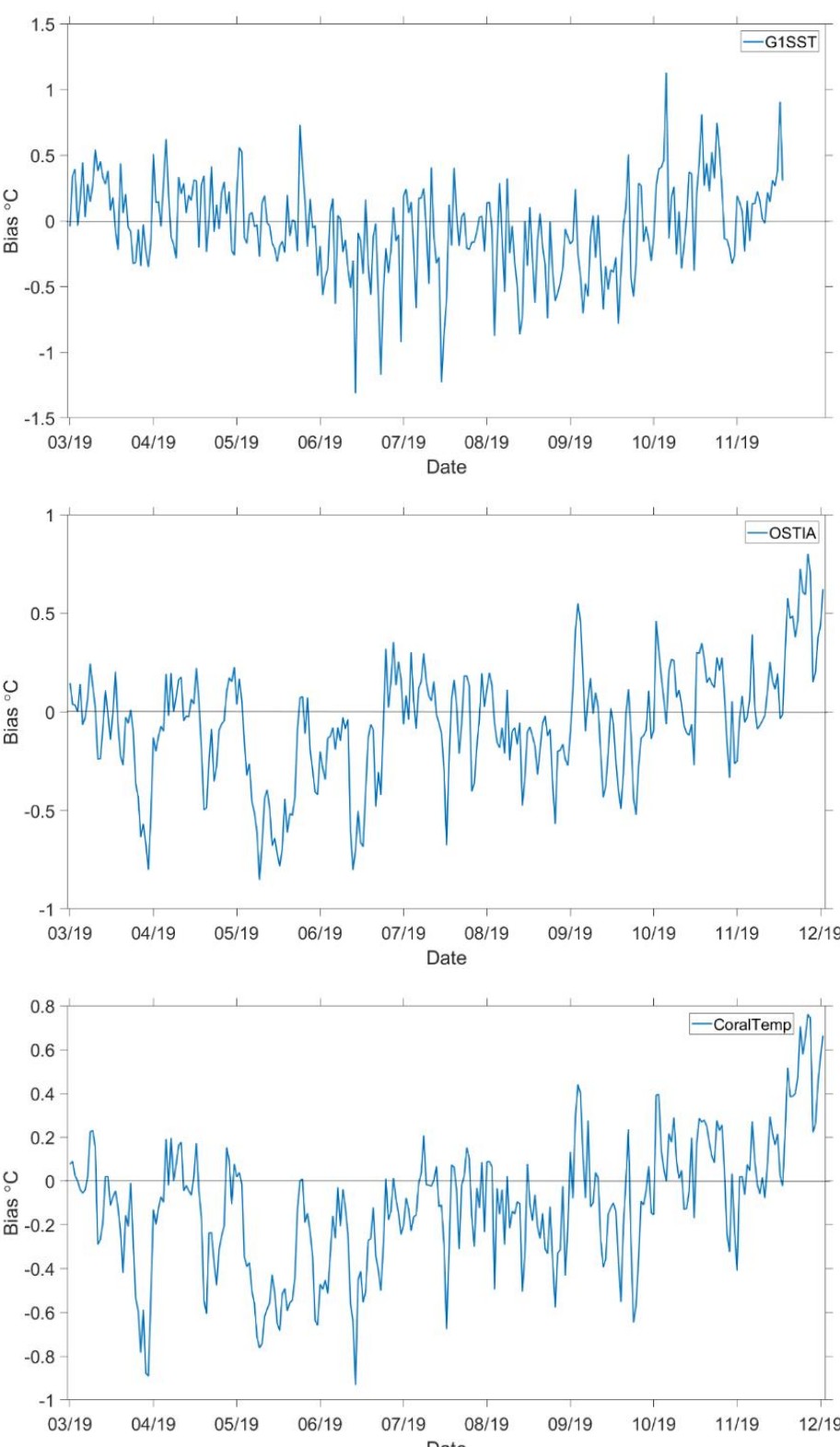

**Figure 7.** Time series of the bias (satellite—in situ) temperatures from 1 March 2019–24 December 2019 for CoralTemp and OSTIA, and from 1 March 2019–8 December 2019 for G1SST for Margarita Reef.

The scatter plots indicated a strong linear relationship between all remote sensing SST datasets and in situ temperature measurements Figures 8–10. Table 9 contains statistical information from the linear regression models.

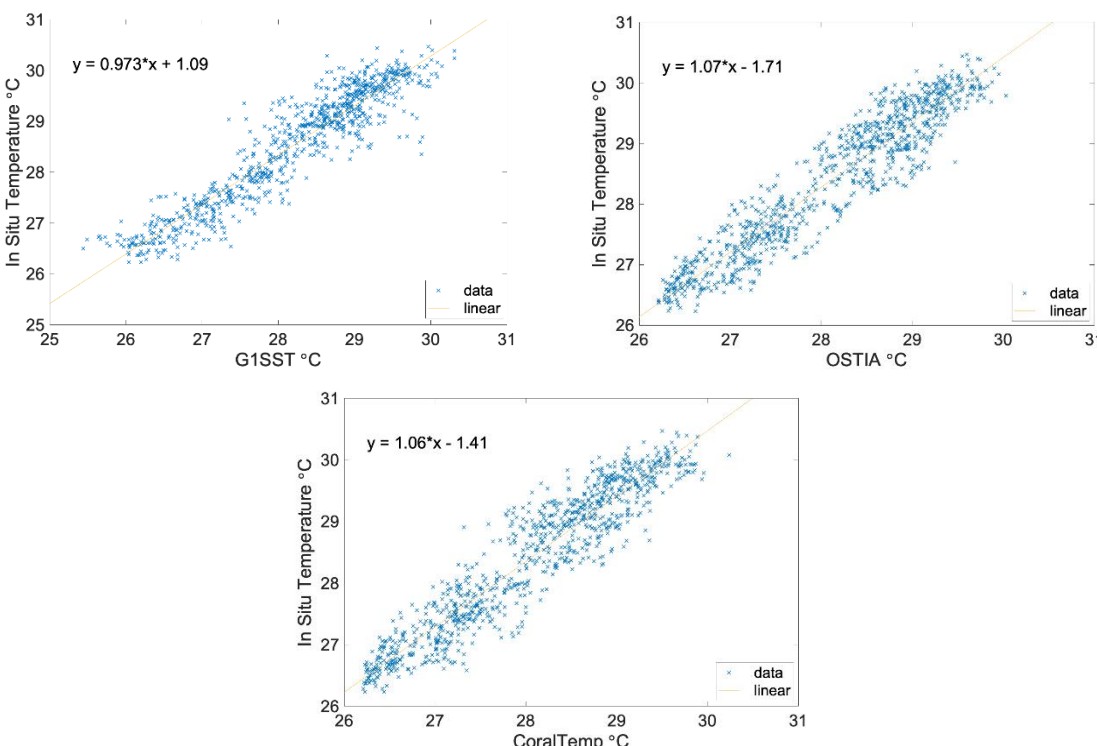

**Figure 8.** Scatter plots with linear regression of the averaged in situ temperature against the different three different remote sensing SST datasets for Cayo Enrique and Cayo Mario.

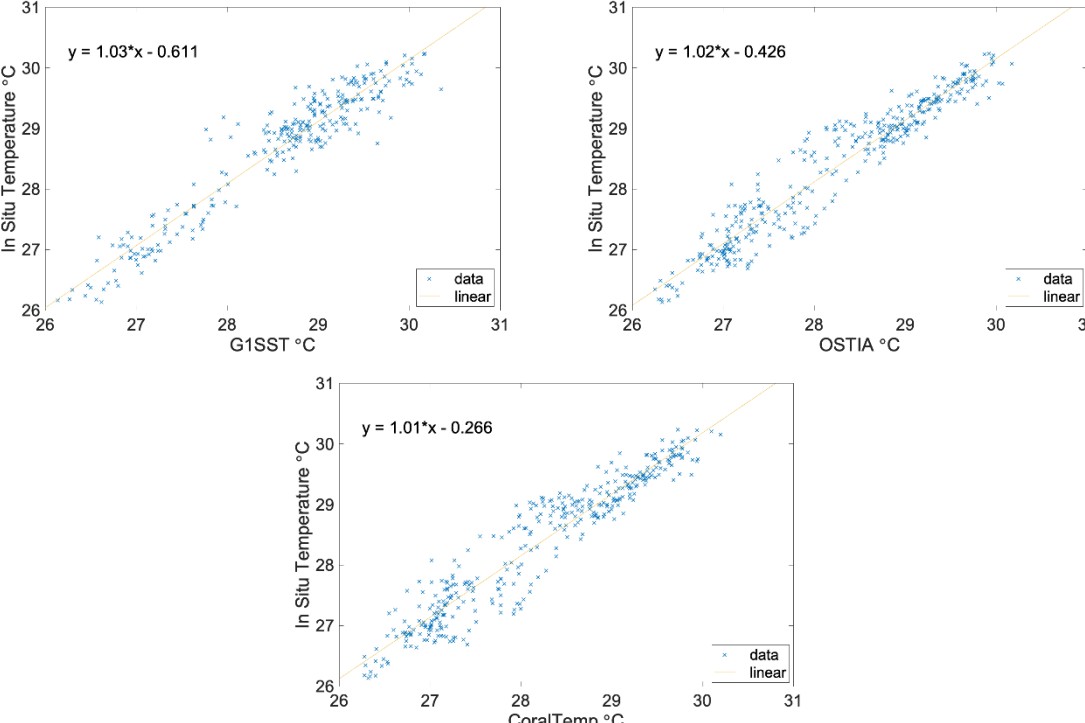

**Figure 9.** Scatter plots with linear regression of the averaged in situ temperature against the different three different remote sensing SST datasets for San Cristobal.

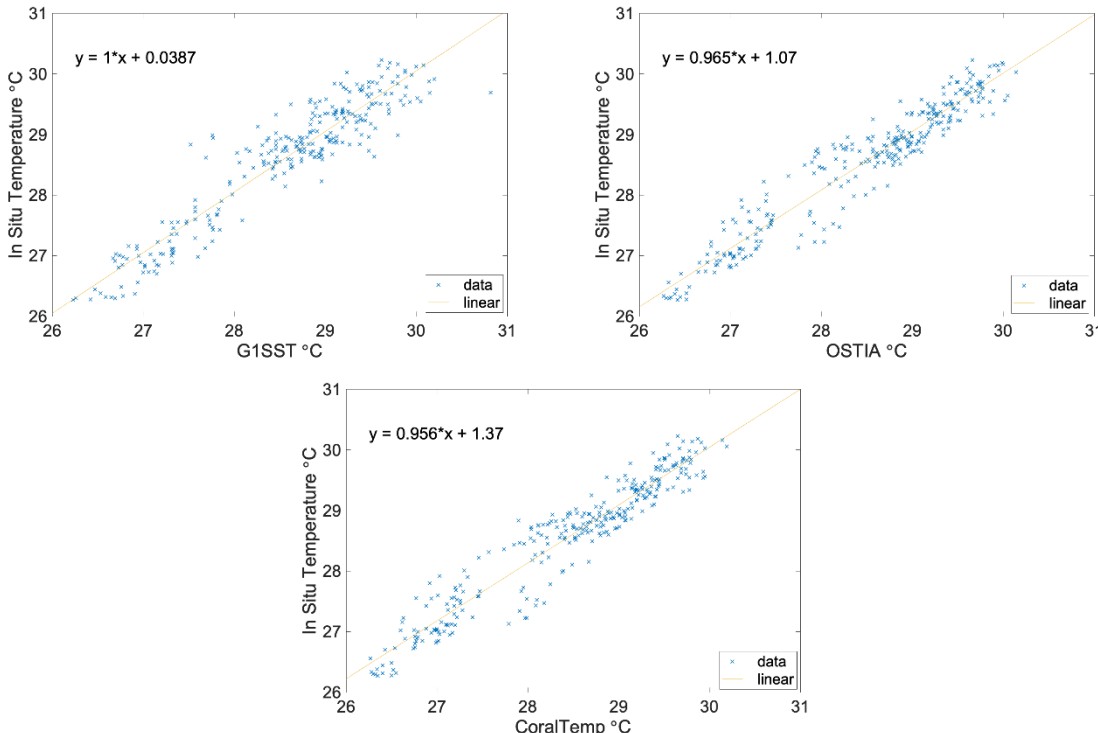

**Figure 10.** Scatter plots with linear regression of the averaged in situ temperature against the different three different remote sensing SST datasets for Margarita Reef.

**Table 9.** Test statistics (root mean square error (RMSE); $R^2$; *p*-value) from the linear regression models, with the three satellite-based SST datasets (CoralTemp = CT; OSTIA; G1SST) as the predictor for the in situ temperature at the three different sites (Cayo E Enrique and Cayo Mario = CE/CM; San Cristobal = SC; Margarita Reef = MR).

| Site | CE/CM | | | SC | | | MR | | |
|---|---|---|---|---|---|---|---|---|---|
| **Statistic** | CT | OSTIA | G1SST | CT | OSTIA | G1SST | CT | OSTIA | G1SST |
| **RMSE** | 0.38 | 0.37 | 0.40 | 0.33 | 0.30 | 0.31 | 0.30 | 0.29 | 0.35 |
| **$R^2$** | 0.89 | 0.89 | 0.87 | 0.91 | 0.92 | 0.91 | 0.92 | 0.92 | 0.88 |
| **_p_-value** | <0.05 | <0.05 | <0.05 | 0.57 | 0.33 | 0.27 | 0.00 | 0.02 | 0.95 |

All remote sensing datasets were negatively biased throughout most of the study period. All scatter plots indicated a strong linear relationship between the satellite-based SST datasets and in situ temperature measurements, suggesting that a linear regression model would be appropriate to estimate coral depth temperatures from the remote sensing data. A linear regression model with CoralTemp, OSTIA, and the G1SST, as the predictor for the in situ temperature measurements yielded an average offset of ~1 °C for all study sites.

## 4. Discussion

The main goal of this research was to assess the suitability of three remote sensing satellite SST datasets, NOAA's CRW CoralTemp, the UK Met Office's OSTIA, and JPL's G1SST dataset, to capture the temperature at the depth of multiple coral reef ecosystems in La Parguera, Puerto Rico. This study discovered that all three of the remote sensing SST datasets evaluated were acceptable surrogates, after offset correction (~1 °C), of the temperature at the depth in the coral reefs at Cayo Enrique, Cayo Mario, San Cristobal, and Margarita Reef.

Overall, all three remote sensing SST products produced a cool bias (satellite—in situ) within their timeframe of data collection, indicating that the satellite SST was underpredicting the actual

temperature at the depth of the corals. For Cayo Enrique and Cayo Mario, the OSTIA SST dataset exhibited the least bias (−0.29 °C, compared to −0.33 °C for G1SST, and −0.35 °C for CoralTemp) for the duration of the study. The G1SST and OSTIA dataset both possessed similar small biases for San Cristobal, at −0.11 °C and −0.12 °C respectively, with CoralTemp yielding −0.16 °C bias for the one-year study. The biases were even smaller for Margarita Reef, with the G1SST bias only −0.05 °C, and the OSTIA SST dataset with a −0.07 °C bias. CoralTemp was found to have a slightly higher cool bias, at −0.12 °C for the ~ten-month study at Margarita Reef. The differences in the satellite SST biases could be attributed to two factors. The first possible explanation is that the study sites were all different depths. The first study sites, Cayo Enrique and Cayo Mario, were overall the deepest sites for the temperature logger deployment (six of the loggers were placed deeper than 6 m, which was the max depth for San Cristobal). San Cristobal was the middle site in terms of depth (5–6 m), with Margarita Reef being the shallowest (3–4 m). The results suggest that the satellite-based SST have a closer temperature matchup with the shallower reefs. This makes sense, because deeper down the water column, the temperature profile drops after the mixed layer (temperature decreases with depth, as it loses availability to sunlight). Another explanation for the bias conflictions between sites is the different time durations of the study for each site. Cayo Enrique and Cayo Mario were the sites with the longest temperature time series (~2 years and 9 months), and they had the relative highest cooler bias offsets for all three satellite SST datasets. Whereas Margarita Reef had the shortest time series (~10 months), and also the smallest biases between the in situ and satellite SST datasets. Longer time series data for all the sites are required to further investigate the biases between the sites and remote sensing SST datasets.

Seasonal patterns were also observed when assessing how representative the satellite-based SST datasets were to the in situ temperature loggers. During the warmer, wet season (June to September), the satellite SSTs often underestimated the temperature at depth recorded by the in situ temperature loggers. This pattern suggests that the satellite SST datasets do not measure the actual temperature at depth when the water is warmer and possibly more stratified [47]. For Cayo Enrique and Cayo Mario, in mid-late August 2017, which is considered to be part of the warm, wet season, a cool bias can be seen in all three remote sensing products. According to NOAA's Climate Review for Puerto Rico 2017, on August 17th and again on the 20th, there was tropical wave activity that created above normal precipitation levels caused by heavy showers and thunderstorms. Then, in September, Hurricanes Irma and Maria caused significant rainfall, destruction, and mixing in the ocean waters. Hurricane Maria was a Category 4 hurricane when it transited directly over the island, passing through on 19–20 September. This hurricane caused the satellite SST data to switch from negative (cool) biases on the 19th (−0.60 °C = CoralTemp; −0.07 °C = G1SST; and −0.05 °C = OSTIA) to positive (warm) biases on the 20th (0.67 °C = CoralTemp; 1.20 °C = G1SST; and 0.78 °C = OSTIA; Figure 11).

According to the in situ data, the ocean temperature dropped ~1.5 °C from the 19th to the 20th, when the hurricane was passing. The warm bias continued until late February 2018, when ocean temperatures began to rise again, and the satellite SST biases returned to a negative (cool) state. The SST data warm biases seen after Hurricane Maria through late February could be attributed to increased water column mixing and seasonal cooler temperatures. Typically, La Parguera experiences low wave and tidal energy, and southeasterly winds from 3.1 to 7.7 m s$^{-1}$ [39,48]. The reversal of the cool bias can be attributed to Hurricane Maria inducing higher seawater mixing for two weeks [49], and then the seasons transitioning to winter, causing temperatures to continue to decrease. From March 2018 to June 2019, SST data have a consistent negative (cool) bias. The lack of intense hurricanes in 2018 and the presence of a La Niña signal (e.g., MEI < −0.5 [50]) could be responsible for the more consistent pattern. NOAA reported lower than average rainfall for Puerto Rico over all of 2018. A weaker El Niño was also present near the end of 2018, continuing into spring 2019. NOAA reported less rainfall than usual for February 2019.

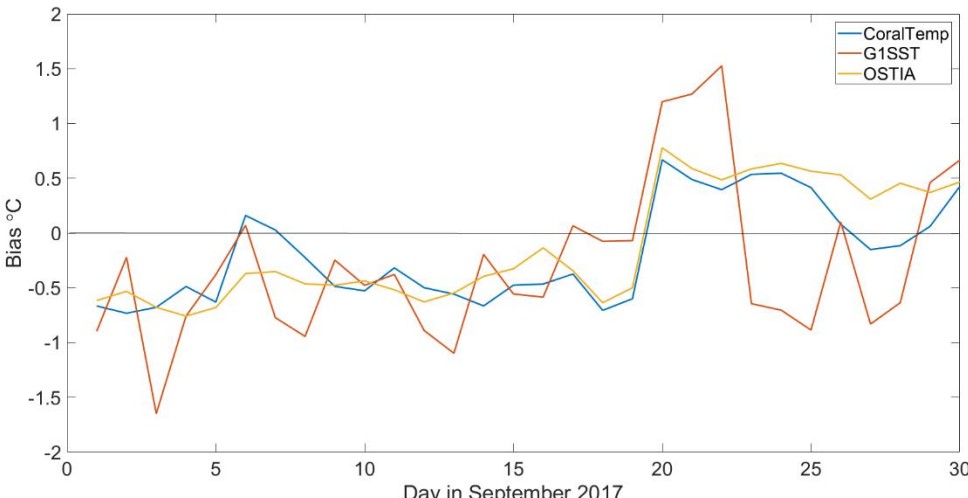

**Figure 11.** Time series of the bias (satellite—in situ) of the three remote sensing datasets for September 2017. All of the remote sensing datasets displayed a cooler bias, until Hurricane Maria transited the area on 19 September 2017, after which the biases became warm.

For San Cristobal, the JPL G1SST dataset yielded the smallest overall bias (−0.11 °C), but all three remote sensing products have high correlations with the in situ temperature measurements (CoralTemp = 0.95, OSTIA = 0.96, G1SST = 0.91). The same is true for Margarita Reef, where all the satellite SST datasets contained high correlations with the in situ data (CoralTemp = 0.95, OSTIA = 0.95, G1SST = 0.91). Observing the time series for all the study sites, there are two surprising temperature dips around December 2019, and January 2020, where the SST datasets are actually over predicting the temperature compared to the in situ loggers, and this was not seen in the previous years for the Cayo Enrique and Cayo Mario sites during these months. Generally, in winter, there was a closer match up observed between the SST data and in situ measurements. A possible explanation for these cool dips in the in situ temperature record that caused the satellites to have a warm bias, could be the magnitude 6.4 earthquake that occurred in southwestern Puerto Rico on 7 January 2020. This region hadn't experienced an earthquake of this magnitude since 1918 (USGS 2020). Scientists believe that the energy released during an earthquake could cause an increase in cold water anomalies, but further research is required to understand how local ocean temperatures change in response to earthquakes [51].

Inevitably, there will be some mismatch between the satellite SST and in situ measurements, because of the spatial scale, as the remote sensing modeled products combine data over a larger area (1–5 km pixel size), and the in situ data are point-based observations. Even though the satellite SST datasets underestimate the actual temperature at the depth of the coral for the majority of this study, the linear regression models with the satellite SST datasets as the predictor for the averaged in situ temperature logger measurements yielded an offset of ~1 °C. Cooler biases were also found in observations between satellite and in situ temperature measurements in previous studies [19,23], suggesting some consistency in bias across geographies. The occurrence of a cooler bias could be due to the ocean surface losing more heat at night to the atmosphere, which the temperature loggers at depth do not experience. Future studies could focus on heat budget analysis in corals in La Parguera, to assess if the coral tissues and reef framework contribute to the observed cool bias.

## 5. Conclusions

We sought to understand correspondence between temperatures observed at coral depth and those estimated for the sea surface by satellite-based remote sensing techniques, and to identify a statistical model capable of correcting bias in the remote sensing data, such that they may more accurately estimate temperatures at coral depths in the near shore zone. Since tropical corals live in coastal, subtidal areas that expose them to a wide range of temperature regimes, it is important to assess

the satellite SST using in situ measurements to gain an accurate understanding of the temperature surrounding each coral reef ecosystem. It is also essential to explore the biases between satellite SST and in situ measurements, because NOAA's CRW uses satellite SST to produce their products, which are used by coral managers to inform them when their reefs might experience bleaching. Overall, a strong positive correlation was observed between all the satellite products and in situ measurements, with no real differences found between logger sites and depth, and this would be expected in such shallow, well-mixed waters. However, a consistent negative (cool) bias was found between the in situ temperature data and satellite SST datasets during the warmer months with a closer match between them during the colder months. The warm season biases for the satellite SST datasets were all around 1 °C, providing a good overall agreement between the satellite SST and the in situ loggers. While other studies have suggested similar biases in different locations, the spatial coherence of systematic biases between surface and reef-depth temperatures has not been fully explored. Expanding this study with a broader network of in situ sensors to simultaneously evaluate more SST pixels is needed, especially around Puerto Rico, and a longer time series is required.

**Author Contributions:** A.M.G. developed the project plan with support from K.C.M. and K.S. in an advisory role. A.M.G. analyzed the satellite data and wrote the manuscript. A.M.G., M.C., K.C.M., and S.D. deployed the in situ sensors, and provided supporting SCUBA fieldwork. K.C.M., K.S. and S.D. provided manuscript edits. R.A.A. and W.J.H. helped with the data collection and manuscript edits. All authors have read and agreed to the published version of the manuscript.

**Funding:** This study is supported and monitored by The National Oceanic and Atmospheric Administration –Cooperative Science Center for Earth System Sciences and Remote Sensing Technologies (NOAA-CESSRST) under the Cooperative Agreement Grant # NA16SEC4810008. The authors would like to thank The City College of New York, NOAA-CESSRST (aka CREST) program and NOAA Office of Education, Educational Partnership Program for full fellowship support for Andrea Gomez. The statements contained within the manuscript/research article are not the opinions of the funding agency or the U.S. government, but reflect the author's opinions. This research is also partly funded by The Doctoral Research Student Grant (DRSG) and the Provost's Pre-dissertation Summer Research Award, both of which come from The CUNY Graduate Center.

**Acknowledgments:** We thank Milton Carlo for his tremendous help with the logger deployment and data collection, and the University of Puerto Rico, Mayagüez Bio-Optical Oceanography, Remote Sensing lab for use of their facilities. We also thank Mark Eakin, and Jacqueline De La Cour from NOAA's Coral Reef Watch for their help editing the manuscript. Finally we thank Aaron Davitt, Brian Lamb, Derek Tesser, Kat Jensen, Michael Brown, and Jessica Rosenqvist from the Ecosystem Science Lab for all their tremendous support and feedback, and Adrian Diaz and Sonia Dagan for their help with Matlab and the statistical analyses. The HOBO in situ temperature data supporting the conclusions can be accessed at the TemperateReefBase Data Portal (https://temperatereefbase.imas.utas.edu.au/static/landing.html).

**Conflicts of Interest:** The authors declare no conflict of interest.

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
