# Peer review of "Comparison of Satellite-Based Sea Surface Temperature to In Situ Observations Surrounding Coral Reefs in La Parguera, Puerto Rico"

_jmse, doi:10.3390/jmse8060453_

Round 1
Reviewer 1 Report
Please to see the attached file. Thank you,

Reviewer 2 Report
There has been a great number of previous studies comparing satellite SST products with in situ temperature measurements. In this sense, the current study does not present any novel approaches. However, most of the previous studies have been performed in the offshore areas. Shallow coastal areas are more complex environments and present difficulties in SST assessment. As such, the accuracy of satellite SST products retrieved from the shallow coastal areas require further assessment. The current study validates the satellite SST products at the depth of coral reefs in La Parguera, Puerto Rico providing the understanding of how reliable the provided SST products are in the temperature assessment in complex coastal environment.
General comments:
- You only emphasize the NOAA CRW dataset in your introduction, while you also used two other SST datasets in your analysis. I would suggest giving a brief introduction also to the other datasets and explain why those datasets were appropriate to use in the present study.
- It is not clear, why in situ temperature sensors were deployed to the varying water depths, when the data from all the deployment depths were averaged in the data analysis. Wasn’t there any temperature variability depending on the deployment depth? Especially in the case of Cayo Enrique and Cayo Mario, where deployment depth varied between 1-13 m? If corals are deeper, is there greater discrepancy between satellite SST data and in situ temperature data?
- In your analysis three SST datasets were used, from which 2 datasets were with 5 km resolution and 1 dataset with 1 km resolution. Throughout the paper you only refer to the 5 km pixel size. For example, in page 4, line 141: “This allowed comparison of three, 5 km pixel cells, rather than one..”, in page 5, line 176: “since all of the data loggers were located within a single CRW SST 5km x 5km grid cell..”, …. It is confusing, as no context was given to the 1 km resolution G1SST dataset.
- Too many tables and graphs are provided throughout the paper. I suggest merging some of the tables and think it through if more complex data presentation is possible.
Specific comments:
Line 24: “In situ data were collected from June 30, 2017 to March 1, 2020 for Cayo Enrique and Cayo Mario…“ No need to provide exact dates of data collection in the abstract. The paper would be easier to follow if this information (deployment time, duration) is provided in the Table, for example as an additional information in the Table 1.
Line 26: “All three satellite-based SST datasets evaluated..” Please define those three datasets (Coraltemp, OSTIA, G1SST) in the abstract, otherwise it is not clear what datasets have been used.
Line 28: “However, all SST datasets exhibited a mean negative (cool) bias..” Maybe it would be more clear, if you say that satellite SST datasets underestimated the temperature when compared with the in situ measurements.
Line 93: “There are no case studies examining the applicability of CRW’s current..” Check the grammar.
Line 98: “… sensors at varying depths across two coral reefs at La Parguera, Puerto Rico.” Shouldn’t there be four coral reefs instead of two? (Cayo Enrique, Cayo Mario, San Cristobal, Margarita Reef).
Figure 1. Text is not visible in the image. The information provided in the text and in the Figure 1 caption does not fully correspond to the information provided in the Table 1. For example, you state that “seven temperature sensors were deployed at Cayo Enrique, and seven temperature sensors at Cayo Mario (n=14 total).” However, in the Table 1 it can be seen that only six sensors were deployed at Cayo Enrique, and six at Cayo Mario (n=12 total).
Line 182: “To facilitate comparison with CRW’s daily global 5km nighttime-only SST product..” CRW SST product was not the only dataset you used in the analysis.
Line 200: “A positive mean bias corresponds to cooler SST data than the in situ data, and a negative bias stands for warmer SST data.” As one of the main results of the current study was that all remote sensing SST datasets displayed negative (cool) biases, where negative bias stands for cooler SST data, then this is a very confusing sentence.
Line 204: “The correlation coefficient from the test varies between +1 and -1..” Please check, can R2 vary between +1 and -1.
Line 220: “For JPL’s Level 4, daily, G1SST dataset, after December 9, 2019 to present, the SST dataset has been reported to produce poor quality SST data.” Should be mentioned that poor data were removed from the analysis.
Line 398: “A possible explanation for the cool..” Can you explain the temperature dip with earthquake if the first dip appeared in December, before the earthquake?
Line 405: “Inevitably, there will be some mismatch..” Can there be any differences between 1 km and 5 km pixel size? Is it possible that smaller pixel size (1 km) has better temperature prediction abilities compared to 5 km pixel size?
Reviewer 3 Report
Dear authors,
The increase in the average temperature of the oceans has direct harmful implications for marine flora and fauna, so the study presented is extremely relevant because it shows that the temperatures obtained by SST techniques are generally lower than the temperatures obtained in situ. And this demonstration has implications for studies based exclusively on SST, which must take into account these temperature differences. This work also demonstrates the importance of in situ work, the data obtained by satellite are convenient but incomplete.
I believe the manuscript can be published in this format.
Best Regards
Author Response
Thank you for the feedback.
Round 2
Reviewer 1 Report
Since the results are of great interest to the community of marine scientists, I accept the work without reservation. Congratulations.
Reviewer 2 Report
I still feel that there are too many graphs/tables in the paper, but I leave it to Editor to decide whether the paper can be accepted at the current state.